# Development of a PET Probe Targeting Bromodomain and Extra-Terminal Proteins for In Vitro and In Vivo Visualization

**DOI:** 10.3390/ph17121670

**Published:** 2024-12-11

**Authors:** Yongle Wang, Yanli Wang, Yulong Xu, Hua Cheng, Tewodros Mulugeta Dagnew, Leyi Kang, Darcy Tocci, Iris Z. Shen, Can Zhang, Changning Wang

**Affiliations:** 1School of Pharmacy, Minzu University of China, Beijing 100081, China; ywang151@mgh.harvard.edu; 2Athinoula A. Martinos Center for Biomedical Imaging, Department of Radiology Massachusetts General Hospital, Harvard Medical School, Charlestown, MA 02129, USA; ywang@mgh.harvard.edu (Y.W.); yulong.xu@mgh.harvard.edu (Y.X.); hcheng9@mgh.harvard.edu (H.C.); tdagnew@mgh.harvard.edu (T.M.D.); lkang5@mgh.harvard.edu (L.K.); darcytocci@brandeis.edu (D.T.);; 3Genetics and Aging Research Unit, Department of Neurology, McCance Center for Brain Health, Massachusetts General Institute for Neurodegenerative Disease, Massachusetts General Hospital, Harvard Medical School, Boston, MA 02129, USA; zhang.can@mgh.harvard.edu

**Keywords:** bromodomains, BRD4 BD1, positron emission tomography, radiotracers, imaging

## Abstract

**Background:** Bromodomain and extra-terminal (BET) proteins are critical regulators of gene transcription, as they recognize acetylated lysine residues. The BD1 bromodomain of BRD4, a member of the BET family, has emerged as a promising therapeutic target for various diseases. This study aimed to develop and evaluate a novel C-11 labeled PET radiotracer, [^11^C]YL10, for imaging the BD1 bromodomain of BRD4 in vivo. **Methods:** [^11^C]YL10 was synthesized and evaluated for its ability to bind to the BD1 bromodomain selectively. PET imaging studies were conducted in mice to assess brain penetration, pharmacokinetics, and selectivity. In vitro autoradiography and blocking experiments were performed to confirm the tracer’s specificity for the BD1 domain. **Results:** [^11^C]YL10 demonstrated good brain penetration, high selectivity for the BD1 bromodomain, and favorable pharmacokinetics in initial PET imaging studies. In vitro autoradiography and blocking experiments confirmed the specific binding of [^11^C]YL10 to the BD1 domain of BRD4, further validating its potential as a targeted radiotracer. **Conclusions:** The development of [^11^C]YL10 provides a new tool for studying BRD4 bromodomains using PET imaging technology. This radiotracer offers potential advancement in the diagnosis and research of neurodegenerative diseases and related disorders involving BRD4 dysregulation.

## 1. Introduction

Bromodomain and extra-terminal (BET) proteins are a key subgroup within the bromodomain-containing protein family, and are essential for regulating gene expression by recognizing acetylated lysine residues on histones and other proteins [1]. This group comprises four main members: BRD2, BRD3, BRD4, and BRDT, each containing two N-terminal bromodomains (BD1 and BD2) and an extra-terminal (ET) domain [2]. BRD4, in particular, plays a crucial role in transcriptional elongation, cell cycle progression, and oncogene expression, partly through its interaction with the positive transcription elongation factor b (P-TEFb) [3]. The capability of BRD4 to influence the transcription of key oncogenes, like MYC, through its association with super-enhancers—large clusters of transcriptional enhancers critical for maintaining cell identity and disease progression, notably in cancer underscores its therapeutic relevance [1]. Among the BET protein family, BRD4 has emerged as a promising therapeutic target for various cancers [2,3,4,5].

Recent investigations have elucidated the pivotal role of BET proteins, particularly BRD4, in the pathophysiology of neuroinflammatory and neurodegenerative disorders. This has been demonstrated through their modulation of inflammatory responses and stress-related pathways. The interaction of BRD4 with super-enhancers, which orchestrate the transcription of genes crucial for neuronal function and inflammatory responses, suggests that BET inhibitors could serve as viable therapeutic agents for conditions such as dementia [6]; notably, small-molecule inhibitors such as JQ1 [7]. Figure 1 shows the efficacy in preclinical models by effectively modulating gene expressions associated with neurodegeneration and cognitive impairment. In contrast, I-BET762 (Figure 1) is primarily effective in attenuating inflammatory processes, highlighting its potential for therapeutic interventions in inflammation-driven neuropathologies [8].

The discovery of JQ1 has greatly enhanced our understanding of BET proteins and their therapeutic relevance [9]. This inhibitor acts by competitively binding to the acetyl-lysine recognition pockets of BET bromodomains, displacing BET proteins from chromatin. This displacement leads to the suppression of MYC transcription and inhibits tumor growth across various cancer models [9,10]. Furthermore, the ability of JQ1 to cross the blood–brain barrier (BBB) facilitates its use in studying BET’s role in neurological disorders such as Alzheimer’s disease, Parkinson’s disease, and schizophrenia [11].

Building upon the success of JQ1, other BET inhibitors like I-BET762 have been developed, exhibiting potent anti-inflammatory activities by disrupting oncogene expression [11,12]. Alongside these pharmacological developments, the inhibitor I-BET858 (Figure 1), a tetrahydroquinoline compound that targets BRD4-BD1, has been recognized for its high specificity and excellent brain permeability. This tracer has become an essential tool for in vivo studies of gene expression modulation within the central nervous system, underscoring its utility in researching neurological disorders [13].

In line with these advancements, our previous research has significantly contributed to the BET field. We developed [^11^C]MS417 (Figure 1), a carbon-11 labeled PET tracer that targets BET bromodomains BD1 and BD2 [14]. It displayed adequate specificity and pharmacokinetics suitable for PET imaging in various animal models, providing crucial insights into BET protein interactions. However, its moderate BBB penetration led us to develop [^18^F]3a and [^18^F]6a (Figure 1), fluorine-18 labeled tracers that enhance brain penetration and targeting specificity [14,15].

These advancements, exemplified by [^18^F]3a and [^18^F]6a, have set new standards in PET imaging by significantly improving our ability to study BET protein functions within the central nervous system. Both tracers have demonstrated high specificity and binding affinity for BET proteins, paving the way for more detailed and accurate imaging of these epigenetic readers in vivo. Our ongoing research, including the development of these tracers, has provided invaluable tools for diagnosing and understanding neurological diseases and disorders associated with dysregulated epigenetic states [15].

In this research, we designed and synthesized a new C-11 labeled PET radiotracer, [^11^C]YL10, which has a dissociation constant (*K_d_*) of 0.48 μM against BRD BD1 protein. Early imaging results indicate that this tracer exhibits strong brain penetration, precise specificity, and suitable pharmacokinetics along with effective brain distribution. Further PET-CT dynamic scanning in mice and autoradiography of brain slices confirmed the tracer’s effective brain penetration and favorable pharmacokinetic profile. Future research aims to evaluate the potential of [^11^C]YL10 in non-human primates (NHPs) and human clinical trials. The demonstrated characteristics of [^11^C]YL10 position it as a potent investigative tool for BET proteins, offering enhanced diagnostic capabilities and therapeutic potential in neurology.

## 2. Results and Discussion

### 2.1. PET Radioligand Design

To date, a variety of BET inhibitors with different chemical structures have been reported in the literature. JQ-1 was the first BET inhibitor to be reported; however, its poor pharmacokinetics have rendered it unsuitable for clinical development. In contrast, other BET inhibitors such as I-BET151 and GSK778 (Figure 1), which contain a quinoline ring structure, have shown improved pharmacokinetic properties [16,17]. The planar structure of the quinoline ring facilitates σ-π interactions with key amino acid residues within BET proteins, such as Pro82 and Leu92, enhancing their binding to BET proteins. Although several BET PET radioligands have been reported, they have not demonstrated sufficient brain uptake in animal models. Therefore, to develop BET PET radioligands with improved brain penetration and domain selectivity, we selected the known domain-selective inhibitor I-BET151, which has an IC_50_ of 0.79 μM against BRD4 (BD1/2) and undertook structural design and modifications.

We made several key structural changes to improve the compound’s brain permeability. As shown in Figure 2, firstly, the pyridine ring was replaced with a non-polar phenyl ring to reduce the molecular polarity, which limits its ability to cross the blood–brain barrier. This modification significantly reduced the compound’s overall polarity, enhancing its permeability. Secondly, replacing the oxygen in dimethyl isoxazole with nitrogen and adding a methyl group created trimethyl pyrazole, which enhanced hydrogen bonding with the target. By reducing the molecular polarity and increasing lipophilicity, we further enhanced brain permeability. Additionally, replacing the oxygen in the imidazolone group with a methyl group decreased the compound’s polarity. This modification may enhance its interaction with hydrophobic targets and improve its ability to cross the blood–brain barrier [18,19,20,21].

### 2.2. Chemical Synthesis of Standard Compound YL10 and Labeling Precursor YL9

Figure 1 illustrates the synthesis of standard compound YL10 via a series of complex reactions [22]. Starting from methyl 2-amino-4-bromo-5-methoxybenzoate (**1**), the ester group is first hydrolyzed under basic conditions to the corresponding acid, resulting in 2-amino-4-bromo-5-methoxybenzoic acid (**2**). Compound **2** is then reacted with nitromethane in water to yield (E)-4-bromo-5-methoxy-2-((2-nitrovinyl)amino)benzoic acid (**3**). The dehydrative cyclization of compound **3** in acetic anhydride produces 7-bromo-6-methoxy-3-nitro-quinolin-4-ol (**4**). Next, compound **4** undergoes a chlorination reaction using phosphorus oxychloride as both the solvent and reagent, forming the intermediate 7-bromo-4-chloro-6-methoxy-3-nitroquinoline (**5**). Compound **5** is then subjected to another nucleophilic substitution reaction with (R)-1-phenylethan-1-amine under basic conditions to yield (R)-7-bromo-6-methoxy-3-nitro-N-(1-phenylethyl)quinolin-4-amine (**6**). Compound **6** is reduced in the presence of iron powder under acidic conditions, producing (R)-7-bromo-6-methoxy-N4-(1-phenylethyl)quinoline-3,4-diamine (**7**). Compound **7** is then coupled with carbonyldiimidazole (CDI) in acetic acid at 100 °C. Through a classic Suzuki reaction, (R)-7-bromo-8-methoxy-2-methyl-1-(1-phenylethyl)-1H-pyrrolo[3,2-c]quinoline (**8**) is reacted with boronic ester to yield the labeling precursor (R)-7-(3,5-dimethyl-1H-pyrazol-4-yl)-8-methoxy-1-(1-phenylethyl)-1,3-dihydro-2H-imidazo[4,5-c]quinolin-2-one (YL9). Finally, YL9 undergoes a methylation reaction with methyl iodide (CH_3_I) and sodium hydride (NaH) in tetrahydrofuran (THF), resulting in the standard compound (R)-8-methoxy-2-methyl-1-(1-phenylethyl)-7-(1,3,5-trimethyl-1H-pyrazol-4-yl)-1H-imidazo[4,5-c]quinoline (YL10), Appendix A.

### 2.3. Ex Vivo Binding Affinity and Selectivity and Physical Properties of YL10

Compound YL10 was tested using the KINOMEscan screening platform and analyzed through quantitative PCR (qPCR), a precise and highly sensitive technique. This method allowed us to accurately measure the interactions between varying concentrations of the tested compounds and bromodomain proteins affixed to a solid support, thereby calculating the dissociation constants (*K_d_*) of the compound with the bromodomains. As shown in Figure 3, the experimental results indicated that YL10 has *K_d_* values of 0.48 μM for BRD4(1) and 1.65 μM for BRD4(2), demonstrating a particularly high affinity towards BRD4(1). This finding prompted the further development of YL10, leading to successful carbon-11 labeling and in vivo positron emission tomography (PET) imaging with [^11^C]YL10.

Physical characteristics such as molecular weight, topological polar surface area (tPSA), and cLog P are commonly employed to estimate the in vivo behavior of radioligand candidates. For instance, a suitable molecular weight, typically below 500, ensures the compound is small enough to pass through the tight junctions between cells, which is critical for crossing the blood–brain barrier. A reasonable topological polar surface area (tPSA), generally recommended to be less than 90, ensures the compound has lower polarity, facilitating easier passage through the lipid bilayer of the blood–brain barrier. Meanwhile, a moderate cLogP value, ideally between −1 and 3, balances the hydrophilicity and lipophilicity, crucial for ensuring effective distribution between the blood and brain tissues. The optimization of these parameters indicates that compound YL10 possesses the physicochemical properties suitable for the development of radioligands (see Table 1), especially demonstrating significant potential for applications involving brain imaging that require penetration through the blood–brain barrier.

### 2.4. Radiosynthesis of [^11^C]YL10

The radiosynthesis of [^11^C]YL10 was performed using a conventional methylation technique. As illustrated in Figure 2, the process began by trapping [^11^C]CH_3_I in anhydrous DMF (300 μL), along with precursor compound YL9 (1.0 mg) and KOH (3.0 mg). The reaction mixture was heated to 100 °C for 3 min. To quench the radioactive reaction, HPLC mobile phase (0.5 mL) was added, and the compound was isolated via reverse-phase semi-preparative HPLC (Agilent Eclipse XDB-C18, 5 mm, 250 mm × 9.4 mm, flow rate = 5.0 mL/min, mobile phase is 0.1% TFA in water/0.1% TFA in acetonitrile, 60/40, *v*/*v*), yielding 21–29% (decay-corrected from trapped [^11^C]CH_3_I). The [^11^C]YL10, identified by a retention time of 10.5 min, was diluted in water (15 mL) and passed through an SPE C-18 cartridge. After rinsing with water (10 mL), the product was eluted using ethanol (1.5 mL). The procedure ensured a radiochemical purity exceeding 95% by HPLC (Agilent Eclipse plus C18, Santa Clara, CA, USA; 3.5 μm, 4.6 × 100 mm, flow rate = 1.0 mL/min, mobile phase = 0.1% TFA in water/0.1% TFA in acetonitrile, gradient method, Appendix A). The identity of [^11^C]YL10 was confirmed by co-injection with a non-radioactive YL10 standard, and the final [^11^C]YL10 solution was prepared in sterile saline (2.7 mL) for subsequent in vivo studies.

### 2.5. Molecular Docking of YL10 with BRD4 BD1

The molecular docking results of compound YL10 with the BRD4 protein reveal critical interactions and its favorable accommodation within the protein’s binding pocket. As shown in Figure 4, the docking analysis indicates that YL10 establishes a strong hydrogen bond with the Lysine (Lys) residue, while also forming stable hydrophobic interactions with Proline (Pro), Threonine (Thr), and Isoleucine (Ile) residues. The three-dimensional structural visualization further corroborates that YL10 is well-integrated into the hydrophobic pocket of BRD4, demonstrating a high binding affinity. The docking score is −7.3755, and reflects the most stable binding conformation among the tested poses. Overall, the high binding affinity and significant molecular interactions of YL10 with BRD4 underscore its potential as a lead compound for further optimization in drug development.

### 2.6. In Vitro Autoradiography Study

We conducted in vitro autoradiographic studies on mouse brain sections using [^11^C]YL10 to assess its binding specificity for BRD4 BD1 [25,26]. Here is a brief overview of the research methods: Mouse brain sections (sagittal plane, 20 μm thickness) were pre-incubated in 50 mM Tris-HCl buffer (pH 7.4) for 20 min. The sections were then exposed to [^11^C]YL10 at a concentration of 37,000 kBq/L in the same buffer. For blocking experiments, unlabeled YL10 (10 μM) was co-incubated with the radiotracer [^11^C]YL10.

After incubation, the brain sections were washed with ice-cold buffer and immersed in chilled deionized water, then allowed to air dry at room temperature. Autoradiographic images were acquired using imaging plates (BAS-MS2025, GE Healthcare, Pittsburgh, PA, USA). Regions of interest (ROIs) were identified through visual observation, and analysis was performed using OptiQuant 4.0 software from PerkinElmer. The results were expressed as photostimulated luminescence per square millimeter (DLU/mm^2^) [27]. Figure 5 shows both baseline and blocked autoradiographic images of mouse sagittal brain sections using [^11^C]YL10. Notably, whole-brain blocking studies indicated a 49.82% reduction in binding, supporting competitive inhibition by YL10, and underscoring the tracer’s specificity for BRD4 binding sites in the brain. Encouraged by these promising in vitro results, we proceeded with in vivo PET studies using [^11^C]YL10.

### 2.7. PET Imaging of [^11^C]YL10 in Mice

Building on the encouraging in vitro data of YL10, we initially conducted dynamic PET-CT imaging studies in mice to evaluate the effectiveness of [^11^C]YL10 as a BET PET imaging probe. Mice were administered [^11^C]YL10 via intravenous injection at doses of 100–150 μCi per mouse, followed by a 60 min dynamic PET scan and a 10 min computed tomography (CT) scan [27]. We evaluated the brain penetration and BRD4 binding selectivity of the [^11^C]YL10 tracer in both the brain and peripheral organs. PET imaging was conducted under baseline and blocking conditions, focusing on BRD4 binding in the brain, and the tracer uptake in the brain, heart, lung, liver, and kidney was compared using Standardized Uptake Values (SUVs) at different time points.

### 2.8. Brain Uptake and Comparison Between Baseline and Blocking

As shown in Figure 6, [^11^C]YL10 demonstrates good blood–brain barrier (BBB) permeability, reaching its maximum standardized uptake value (SUV) 5 min post-injection. The time-activity curves (TACs) indicate that [^11^C]YL10 rapidly crosses the BBB and maintains stable brain radioactivity levels throughout the entire 60 min scanning period. At baseline, the [^11^C]YL10 tracer exhibits moderate brain uptake, peaking at an SUV of 0.5–0.6 at early time points (e.g., 5 min). This suggests that the tracer successfully crosses the BBB and persistently binds to BRD4 BD1 in the brain.

In the blocking group, a rapid increase in early brain SUV is observed, peaking at 5 min. This phenomenon is likely due to the unlabeled YL10 occupying the binding sites, leading to higher blood concentrations of [^11^C]YL10. The elevated blood activity contributes to early brain signals via vascular spillover effects, resulting in an apparent increase in SUV in the blocking group. Importantly, the TACs for the blocking group show a rapid washout of radioactivity, with brain SUV levels decreasing by approximately 69% at 60 min post-injection (Figure 6B). This rapid decline directly demonstrates the successful blocking of specific binding sites by unlabeled YL10. To minimize the contribution of blood-derived signals and better isolate the tracer-specific binding, normalization by the maximum radioactivity uptake in the blood pool was performed. After this correction, a significant blocking effect is observed in mice pretreated with unlabeled YL10 (Figure 6D), making the results more distinct and clearly demonstrating that [^11^C]YL10 has high specific binding to BRD4 BD1 in the brain.

### 2.9. Regional Brain Uptake of [^11^C]YL10 in Mice

The regional uptake of [^11^C]YL10 in mouse brains was examined across several functional areas. Specifically, the cortex, cerebellum, thalamus, hypothalamus, striatum, hippocampus, and amygdala were selected for analysis. The data from these regions were collected using the Ma–Benveniste–Mirrione VOI atlas within the fusion module of PMOD software (PMOD 4.01, PMOD Technologies Ltd., Zurich, Switzerland) [28]. Figure 7 displays the time-activity curves (TACs) of [^11^C]YL10 uptake in these brain regions over 60 min post-injection. Initially, all regions showed rapid uptake within the first 5–10 min, followed by a gradual decrease or stabilization. Notably, the striatum, cortex, and brain stem exhibited slightly higher initial uptake compared to other regions like the cerebellum and thalamus, which had lower levels. Overall, [^11^C]YL10 demonstrated consistent distribution across the examined brain regions throughout the duration of the scan.

### 2.10. Peripheral Organ SUV Analysis

Using standardized uptake values (SUV), we evaluated the biodistribution and uptake of [^11^C]YL10 in target organs. Initially, the systemic biodistribution of [^11^C]YL10 in mice was assessed. As shown in Figure 8, four time points post-injection (5, 15, 30, and 60 min) were selected to analyze the uptake, distribution, and clearance of the radioligand in major organs. [^11^C]YL10 exhibited widespread distribution in blood-rich organs such as the liver and kidneys. However, the uptake in the heart and lungs was relatively low. In the liver, radioactivity levels increased rapidly after injection, followed by a gradual decrease, while in the kidneys, radiotracer uptake steadily rose, indicating metabolic processing through the hepatobiliary and urinary systems [29].

## 3. Materials and Methods

### Experimental Section

General Information: All reactions were performed under a nitrogen (N_2_) atmosphere with dry solvents in anhydrous conditions unless otherwise stated. Commercially available reagents were used directly without additional purification. Reaction progress was monitored using liquid chromatography-mass spectrometry (LCMS). Nuclear magnetic resonance (NMR) spectra were recorded on a 400 MHz instrument, with chemical shifts (δ) reported in parts per million (ppm) relative to tetramethylsilane (TMS) as the internal reference and coupling constants (J) expressed in hertz (Hz).

Radiochemistry: The study was conducted following our previously established method. [^11^C]CO_2_ was generated through the 14N (p, α)^11^C reaction with 2.5% oxygen and 11 MeV protons using a Siemens Eclipse cyclotron in nitrogen, and then captured on molecular sieves by a TRACERlab FX-MeI synthesizer (General Electric, Chicago, IL, USA). [^11^C]CH_4_ was produced by reducing [^11^C]CO_2_ in the presence of Ni/hydrogen at 350 °C, after which it was recirculated through an oven containing iodine (I_2_) to produce [^11^C]CH_3_I via a radical reaction.

Ethical Approval and Animal Preparation: The study received the necessary ethical approval from the Institutional Animal Care and Use Committee (IACUC) under the oversight of the Subcommittee on Research Animal Care (SRAC) at Massachusetts General Hospital (MGH). The experiments were conducted using eight male C57BL6 mice aged five months. Each mouse was prepared for imaging by dissolving YL10 into a 1 mg/mL concentration using a mixture composed of 10% DMSO, 10% Tween 80, and 80% saline as the vehicle.

Anesthesia and Imaging Protocol: During the PET/CT scans, mice were anesthetized with 1–1.5% isoflurane, maintaining consistent anesthesia levels for the duration of imaging. The imaging process involved placing each mouse within a Triumph PET/CT scanner (Gamma Medica, Northridge, CA, USA). We administered the radiotracer [^11^C]YL10 intravenously with doses ranging from 8251 to 8547 KBq. Initial pretreatments included a five-minute pre-administration of YL10 4.0 mg/kg dosages for one mouse each, or a vehicle administration for another, all via lateral tail vein catheter.

Data Acquisition and Image Processing: After radiotracer injection, a dynamic 60 min PET scan was conducted, immediately followed by a CT scan. PET data were reconstructed using the 3D Maximum Likelihood Expectation Maximization (3D-MLEM) method, achieving a spatial resolution of 1 mm at full width at half maximum. The imaging data were processed using Technologies software (PMOD 4.01, PMOD Technologies Ltd., Zurich, Switzerland), which handled PET and CT images in DICOM format and facilitated their alignment with a standard brain atlas. For detailed analysis, volumes of interest (VOIs) were defined as elliptical regions in both the body and brain, integrating PET and CT data. Time-activity curves (TACs) were calculated, and standardized uptake values (SUVs) were derived to quantify tracer activity per unit volume, corrected for attenuation.

## 4. Conclusions

In conclusion, we developed and successfully evaluated a novel BRD4 BD1 PET radioligand, [^11^C]YL10, in vivo and in vitro. We synthesized both the unlabeled YL10 and the radiolabeled precursor YL9. In vitro studies demonstrated that YL10 exhibits moderate binding affinity and selectivity towards BRD4 BD1. Utilizing the classic [^11^C]CH_3_I methylation radio-synthesis technique, we produced [^11^C]YL10 with a good radiochemical yield (RCY). This tracer displayed good brain permeability and binding to BRD4 BD1. Blocking experiments showed that pretreatment with unlabeled YL10 reduced [^11^C]YL10 uptake in the mouse brain, confirming its specificity and stability. Overall, our results indicate that [^11^C]YL10 serves as an effective PET radioligand for studying BRD4 BD1 in the brain.

To further enhance the specificity of the tracer for BRD4 BD1 in the brain, future research should focus on minimizing non-specific binding and further optimizing the molecular design of the tracer to enhance its selectivity for BRD4 BD1. Additionally, continuing studies on the metabolism and clearance of the tracer in peripheral organs will help improve its overall pharmacokinetic properties, ensuring a more accurate assessment of BRD4 BD1 binding in the brain. This study lays the groundwork for further optimization of the [^11^C]YL10 tracer, providing key insights for its application in brain BRD4 imaging.

## Data Availability

The data supporting the findings of this study are included within the article.

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
