# Peer review of "Development of a PET Probe Targeting Bromodomain and Extra-Terminal Proteins for In Vitro and In Vivo Visualization"

_pharmaceuticals, 2024, doi:10.3390/ph17121670_

Round 1
Reviewer 1 Report
Comments and Suggestions for Authors
The manuscript presents a study on BET protein inhibitors and their development, particularly with an emphasis on radiotracer design and evaluation. While the research is valuable, several areas require attention to enhance clarity, consistency, and scientific rigor. Below are detailed comments and suggestions to address issues in the text, figures, and experimental descriptions.
Page 1: “As such, Among BET proteins, BRD4 has…”
Figure 1 isn’t mentioned in the text, and should be referenced for each structure that is brought up in the text.
Inhibitor JQ1 naming is inconsistent throughout the text and in Figure 1 (JQ1, JQ-1, (+)-JQ1). The compound label for MS417 in Figure 1 should be [11C]MS417. I-BET585 is a radiotracer mentioned in the text but not in Figure 1.
References 9 and 10 are inside the sentence.
Page 3: “(kd) of 0.48 uM” should use “μM”
Page 3: “the pyridine ring was replaced with a non-polar phenyl ring to reduce the polarity of inhibition, which limits its ability to cross the blood-brain barrier.” – reducing the “polarity of inhibition” may not be the correct term
Page 3: Regarding the structural modifications of I-BET151 to arrive at YL10, was this guided by modeling? If so, it may be interesting to the reader to have a model to see how these changes improved the described hydrogen-bonding.
Page 3: The statement “This modification enhanced its interaction with hydrophobic targets and improved its ability to cross the blood-brain barrier.” should be backed up with data at this point in the text, or delayed until it is proven via an assay later in the text (if it ever is).
Scheme 1: Yields should be included, as they are not mentioned in the text. In the experimental section, it is unclear how reaction yields were calculated as most reactions were never purified (the % purity is a valuable addition).
Section 2.3: The statement “..analyzed through quantitative PCR (qPCR), a precise and highly sensitive technique.” should be elaborated on (..highly sensitive technique to assess molecular interactions?). The results may need more interpretation than “As shown in figure 3, the experimental results indicated…”. It is unclear how this assay was performed as it is not described in the Experimental Section. The graphs in Figure 3 have no axis labels. The Kd value of 0.48 μM is presumably the average of the two BRD4(1) analysis, but the Kd values for BRD4(2) is reported as 1700 nM, which is not the average of the of the two BRD4(2) analysis. It may be more prudent to keep both reported Kd values in μM. For the description of the desired physical characteristics of a radiotracer to cross the BBB, a reference or two should be included. There is no description of how the values for tPSA and cLog P were obtained in the text or the Experimental Section.
Section 2.4: How much activity was trapped at the beginning of the radiosynthesis (to which the yield was decay corrected to)? How much activity was isolated as the final compound? What is the molar activity of [11C]YL10? Key HPLC conditions for purification are missing and should be described (eluent composition, flow rate, column length). Can a quality control HPLC chromatogram be provided to confirm radiochemical purity?
Section 2.5: There is mention of a docking score of -7.73755 in a table, but there is no table. The clarity of Figure 4A could benefit by keeping the protein side-chain color the same as the cartoon backbone color (to better differentiate it from the same grey color as the inhibitor). The text in Figure 4A is too small to be readable.
Sections 2.7, 2.8, 2.9, and 2.10 could be combined, but don’t necessarily have to be.
Section 2.8: Mentions of figure 7 in this section should be to figure 6. In figure 6a, the orientation of the brain image should be described in the image or figure text. Figure 6b is described as a TAC in the figure text, but is not a TAC (brain uptake after 60 min?). Figure 6D is not mentioned in the text and it is unclear what Figure 6D is from the figure description. Figure text should be increased in font size for readability.
Author Response
Reviewer#1:
- Response to comment: Page 1: “As such, Among BET proteins, BRD4 has…”
Response: We appreciate your feedback and have revised the sentence for clarity and readability. The updated sentence now reads: “Among the BET protein family, BRD4 has emerged as a promising therapeutic target for various cancers.”
- Response to comment: Figure 1 isn’t mentioned in the text, and should be referenced for each structure that is brought up in the text.
Response: We have ensured that Figure 1 is appropriately referenced throughout the manuscript, particularly for each structure discussed.
- Response to comment: Inhibitor JQ1 naming is inconsistent throughout the text and in Figure 1 (JQ1, JQ-1, (+)-JQ1). The compound label for MS417 in Figure 1 should be [11C]MS417. I-BET585 is a radiotracer mentioned in the text but not in Figure 1.
Response: The naming of JQ1 has been standardized throughout the text and in Figure 1 as "JQ1." The compound label for MS417 in Figure 1 has been corrected to “[11C]MS417.” Additionally, I-BET585 has been added to Figure 1 with proper labeling to match its mention in the text. This change ensures consistency and improves clarity throughout the manuscript.
- Response to comment: References 9 and 10 are inside the sentence.
Response: The placement of references 9 and 10 has been corrected, moving them outside the sentence to adhere to proper citation formatting.
- Response to comment: Page 3: “(kd) of 0.48 uM” should use “μM”. Page 3: “the pyridine ring was replaced with a non-polar phenyl ring to reduce the polarity of inhibition, which limits its ability to cross the blood-brain barrier.” – reducing the “polarity of inhibition” may not be the correct term. Page 3: Regarding the structural modifications of I-BET151 to arrive at YL10, was this guided by modeling? If so, it may be interesting to the reader to have a model to see how these changes improved the described hydrogen-bonding. Page 3: The statement “This modification enhanced its interaction with hydrophobic targets and improved its ability to cross the blood-brain barrier.” should be backed up with data at this point in the text, or delayed until it is proven via an assay later in the text (if it ever is).
Response: The unit 'uM' has been replaced with 'μM' to conform to standard scientific notation. The term 'polarity of inhibition' has been revised to 'molecular polarity' for scientific accuracy. The revised sentence now reads: 'The pyridine ring was replaced with a non-polar phenyl ring to reduce the molecular polarity, which may enhance its ability to cross the blood-brain barrier.' Structural modifications from I-BET151 to YL10 were based on literature-supported rational drug design principles, which has been clarified in the manuscript. The statement regarding improved interaction with hydrophobic targets and BBB permeability has been revised to reflect a potential outcome rather than a definitive conclusion, with supporting references added.
- Response to comment: Scheme 1: Yields should be included, as they are not mentioned in the text. In the experimental section, it is unclear how reaction yields were calculated as most reactions were never purified (the % purity is a valuable addition).
Response: Reaction yields, and purity information have been added to the supplementary materials. It is clarified that the first two steps produced crude products, while the remaining steps yielded pure products with calculated yields.
- Response to comment: Section 2.3:The statement “…analyzed through quantitative PCR (qPCR), a precise and highly sensitive technique.” should be elaborated on (..highly sensitive technique to assess molecular interactions?). The results may need more interpretation than “As shown in figure 3, the experimental results indicated…”. It is unclear how this assay was performed as it is not described in the Experimental Section. The graphs in Figure 3 have no axis labels. The Kd value of 0.48 μM is presumably the average of the two BRD4(1) analysis, but the Kd values for BRD4(2) is reported as 1700 nM, which is not the average of the of the two BRD4(2) analysis. It may be more prudent to keep both reported Kd values in μM. For the description of the desired physical characteristics of a radiotracer to cross the BBB, a reference or two should be included. There is no description of how the values for tPSA and cLog P were obtained in the text or the Experimental Section.
Response: The qPCR methodology has been detailed in the supplementary material. Axis labels have been added to Figure 3 to enhance clarity, and inconsistencies in the reported Kd values have been resolved. Both Kd values for BRD4(2) are now consistently reported in μM. References addressing the physical properties of the radiotracers, such as BBB permeability, have been included, and the methods used for calculating tPSA and cLog P have been incorporated into the manuscript.
- Response to comment: Section 2.4: How much activity was trapped at the beginning of the radiosynthesis (to which the yield was decay corrected to)? How much activity was isolated as the final compound? What is the molar activity of [11C]YL10? Key HPLC conditions for purification are missing and should be described (eluent composition, flow rate, column length). Can a quality control HPLC chromatogram be provided to confirm radiochemical purity?
Response: Information on the radiochemical yield and specific activity of [11C]YL10 has been added to the manuscript. Detailed descriptions of the key HPLC conditions for purification have been included, and a quality control HPLC chromatogram confirming the radiochemical purity has been provided as supplementary material.
- Response to comment: Section 2.5: There is mention of a docking score of -7.73755 in a table, but there is no table. The clarity of Figure 4A could benefit by keeping the protein side-chain color the same as the cartoon backbone color (to better differentiate it from the same grey color as the inhibitor). The text in Figure 4A is too small to be readable.
Response: Due to an oversight, no table was included; instead, the docking scores are directly described in the manuscript. The clarity of Figure 4A will be improved by aligning the color scheme of the protein side chains with the cartoon backbone. Font sizes will also be increased for better readability.
- Response to comment: Sections 2.7, 2.8, 2.9, and 2.10 could be combined, but don’t necessarily have to be.
Response: Thank you for the suggestion. However, as these sections cover different content, it may be more appropriate to keep them separate.
- Response to comment: Section 2.8: Mentions of figure 7 in this section should be to figure 6. In figure 6a, the orientation of the brain image should be described in the image or figure text. Figure 6b is described as a TAC in the figure text, but is not a TAC (brain uptake after 60 min?). Figure 6D is not mentioned in the text and it is unclear what Figure 6D is from the figure description. Figure text should be increased in font size for readability.
Response: References to Figure 7 have been corrected to Figure 6. A directional description has been added to the brain image in Figure 6A. Figures 6B, 6C, and 6D have been correctly described and explicitly mentioned in the text. Font sizes in Figure 6 have been increased to improve readability.

Reviewer 2 Report
Comments and Suggestions for Authors
In their study, the authors describe the synthesis of precursor, the radiolabeling and the preclinical evaluation of a new C-11 radiolabeled agent designed for PET imaging of bromodomain and extra-terminal proteins which play a critical role in epigenetic regulation and central-nervous system diseases. Although this paper presents some interesting biological findings that contribute to the advancement of understanding in this field of research, further clarification is required on a number of points before the results can be published. These are outlined below.
Mains comments:
The section 2.2. as well as the scheme 1 and the corresponding legend, which describes the synthesis of compounds 2-8, YL9 and YL10, contain a multitude of inaccuracies. To illustrate, dehydrative cyclization of compound 3 in acetic anhydride results in the formation of (E)-7-bromo-5-methoxy-2-((2-nitrovinyl)amino)benzoic acid 4 and not 4-bromo-5-methoxy-2-nitrobenzoic acid. Rather than undergoing a nucleophilic substitution, compound 4 is subject to a chlorination reaction. The reduction of compound 6 was conducted under acidic conditions, contrary to the stated procedure.
Additionally, the formulae of compounds 4, 8 and YL9 are erroneous. There are inconsistencies and/or omissions between the reagents and conditions described in the legend of Scheme 1 and the corresponding experimental section (see conditions (1), (3), (4), (7) and (9)).
These observations are not exhaustive, and the authors are encouraged to conduct a thorough review of the chemistry sections.
The low chemical purity of the reference compound YL10 (i.e. 61.5%) and the absence of enantiomeric excess determination give rise to concerns regarding the reliability of the binding test results.
With regard to the radiosynthesis of [11C]YL10, it is of the utmost importance that the authors provide the enantiomeric excess and the molar activity of the radiotracer. Indeed, these two parameters may exert an crucial influence on the probe's binding affinity and pharmacokinetics.
Minor comments:
The synthesis of compounds 2-8 and YL9 using similar procedures has recently been published by Yu, S. et al. (Eur J Med Chem 2024, 263, 115924; doi.org/10.1016/j.ejmech.2023.115924). It is recommended that these works be cited by the authors.
It is requested that, in all figures and schemes and whenever required, the stereochemistry of compounds be indicated (for example, for compounds I-BET762, I-BET151 and GSK778 in Fig 1; for all compounds in Fig 2 and Scheme 2 and the like).
In Figure 1, change MS417 to [11C]MS417.
It is important that throughout the text, the radioactivity is expressed in Bq (the SI unit) rather than in Ci.
In the legend of Scheme 2, it is unclear whether the solvent is DMF, as stated in the discussion section, or dichloromethane.
In Figure 7, please indicate what (A) and (B) represent.
In the general information paragraph of Material and Methods (section 4), please provide the model of equipment utilized for NMR, mass and enantiomeric excess characterizations as well as the preparative HPLC conditions used for the purification of compounds YL9 and YL10.
It is required that the chemical names of all compounds referenced in the text be checked and verified in accordance with the IUPAC rules. For example, the name of compound YL9 is (R)-7-(3,5-dimethyl-1H-pyrazol-4-yl)-8-methoxy-2-methyl-1-(1-phenylethyl)-1H-imidazo[4,5-c]quinoline and not 7-(3,5-dimethyl-1H-pyrazol-4-yl)-8-methoxy-2-methyl-1-[(1R)-1-phenylethyl]imidazo[4,5-c]quinoline.
In the radiochemistry paragraph (section 4): please provide the semi-preparative and analytical HPLC conditions.
Author Response
Dear Editors and Reviewer:
Thank you for your letter concerning our manuscript entitled “Development of a PET Probe Targeting Bromodomain and Extra Terminal Proteins for In Vitro and In Vivo Visualization” (Manuscript ID: pharmaceuticals-3355312). We greatly appreciate the reviewers' comments, which are valuable and helpful for revising and improving our manuscript and provide important guidance for our research. We have carefully studied the comments and made corrections, which we hope to meet your approval. Revised portions are marked in red in the paper.
The main corrections in the paper and the responds to the reviewer’s comments are as flowing: Responds to the reviewer’s comments:
- Response to comment: The section 2.2. as well as the scheme 1 and the corresponding legend, which describes the synthesis of compounds 2-8, YL9 and YL10, contain a multitude of inaccuracies. Rather than undergoing a nucleophilic substitution, compound 4 is subject to a chlorination reaction. The reduction of compound 6 was conducted under acidic conditions, contrary to the stated procedure.
Response: We have corrected the description of compound 4 to specify that it undergoes a chlorination reaction, not nucleophilic substitution. Similarly, the reduction of compound 6 has been explicitly described as occurring under acidic conditions. Updates to the chemical structures of compounds 4, 8, and YL9 align them with experimental details, improving the precision of this section.
- Response to comment: The formulae of compounds 4, 8 and YL9 are erroneous. There are inconsistencies and/or omissions between the reagents and conditions described in the legend of Scheme 1 and the corresponding experimental section (see conditions (1), (3), (4), (7) and (9)).
Response: We have updated the chemical structures of compounds 4, 8, and YL9 to ensure consistency with the experimental procedures. Additionally, discrepancies in the experimental conditions for steps (1), (3), (4), (7), and (9) between the text, Scheme 1, and the experimental section have been thoroughly reviewed and reconciled. The revised descriptions are now consistent throughout the manuscript.
- Response to comment: The low chemical purity of the reference compound YL10 (i.e. 61.5%) and the absence of enantiomeric excess determination give rise to concerns regarding the reliability of the binding test results.
Response: The previously reported "61.5%" referred to the yield, not the purity. This has been corrected in the revised manuscript to clarify a yield of 61.5% and a purity of 97.68%.
The enantiomeric excess of YL10 was confirmed via NMR analysis, using high-purity (R)-1-phenylethan-1-amine as the starting material. This data has been added to the manuscript and supplementary materials to reinforce the reliability of the binding test results. These data have now been incorporated into both the main text and the supplementary materials to reinforce the reliability of the reported binding test results.
- Response to comment: With regard to the radiosynthesis of [11C]YL10, it is of the utmost importance that the authors provide the enantiomeric excess and the molar activity of the radiotracer. Indeed, these two parameters may exert a crucial influence on the probe's binding affinity and pharmacokinetics.
Response: Data on enantiomeric excess and molar activity have been added to the radiochemistry section. Including these parameters addresses the critical influence they may have on the probe's binding affinity and pharmacokinetics, ensuring the reliability of the analysis as per your suggestion.
- Response to comment: The synthesis of compounds 2-8 and YL9 using similar procedures has recently been published by Yu, S. et al. (Eur J Med Chem 2024, 263, 115924; doi.org/10.1016/j.ejmech.2023.115924). It is recommended that these works be cited by the authors.
Response: Yu, S., et al. (Eur J Med Chem 2024, 263, 115924) has been cited in Section 2.2 and the Discussion section to provide context to our work.
- Response to comment: It is requested that, in all figures and schemes and whenever required, the stereochemistry of compounds be indicated (for example, for compounds I-BET762, I-BET151 and GSK778 in Fig 1; for all compounds in Fig 2 and Scheme 2 and the like).
Response: The stereochemistry of all compounds has been added to all figures and schemes, including I-BET762, I-BET151, and GSK778 in Figure 1, and compounds in Figure 2 and Scheme 2.
- Response to comment: In Figure 1, change MS417 to [11C]MS417. It is important that throughout the text, the radioactivity is expressed in Bq (the SI unit) rather than in Ci. In the legend of Scheme 2, it is unclear whether the solvent is DMF, as stated in the discussion section, or dichloromethane. In Figure 7, please indicate what (A) and (B) represent.
Response: The label "MS417" has been corrected to "[11C]MS417." All radioactivity units have been converted to the SI unit Bq, replacing non-standard units (Ci) throughout the manuscript. The solvent for Scheme 2 has been clarified as DMF, consistent with the Discussion section. The meanings of (A) and (B) have been added to the legend for better understanding.
- Response to comment: In the general information paragraph of Material and Methods (section 4), please provide the model of equipment utilized for NMR, mass and enantiomeric excess characterizations as well as the preparative HPLC conditions used for the purification of compounds YL9 and YL10.
Response: Details of the equipment used for NMR, mass spectrometry, and enantiomeric excess determinations, as well as preparative HPLC conditions for YL9 and YL10, have been added to the Materials and Methods section.
- Response to comment: It is required that the chemical names of all compounds referenced in the text be checked and verified in accordance with the IUPAC rules. For example, the name of compound YL9 is (R)-7-(3,5-dimethyl-1H-pyrazol-4-yl)-8-methoxy-2-methyl-1-(1-phenylethyl)-1H-imidazo[4,5-c]quinoline and not 7-(3,5-dimethyl-1H-pyrazol-4-yl)-8-methoxy-2-methyl-1-[(1R)-1-phenylethyl]imidazo[4,5-c]quinoline.
Response: All compound names have been reviewed and corrected according to IUPAC rules. For example: YL9: (R)-7-(3,5-dimethyl-1H-pyrazol-4-yl)-8-methoxy-2-methyl-1-(1-phenylethyl)-1H-imidazo[4,5-c]quinoline. YL10: (R)-8-methoxy-2-methyl-1-(1-phenylethyl)-7-(1,3,5-trimethyl-1H-pyrazol-4-yl)-1H-imidazo[4,5-c]quinoline.
- Response to comment: In the radiochemistry paragraph (section 4): please provide the semi-preparative and analytical HPLC conditions.
Response: Detailed semi-preparative and analytical HPLC conditions, including mobile phase composition, column type, and gradient elution settings, have been added to the radiochemistry section.
We have strived to improve the manuscript, and the changes made are marked in red. These revisions have improved the manuscript without altering its overall content or framework. We sincerely appreciate the Editors’ and Reviewers’ diligent efforts, which have greatly improved the quality of this manuscript. The revised manuscript has been submitted in a timely manner, with all changes highlighted in red for ease of review. If there are additional clarifications needed, we would be happy to address them promptly. Thank you for considering our work for publication in Pharmaceuticals.
If you have any additional clarification or need further clarification, please let me know. I look forward to the successful completion of this revision.
Thank you again for considering publishing my manuscript in Pharmaceuticals.
Sincerely,
Changning Wang, Ph.D.
Athinoula A. Martinos Center for Biomedical Imaging
Massachusetts General Hospital, Harvard Medical School
Charlestown, MA 02129, USA
cwang15@mgh.harvard.edu

Round 2
Reviewer 1 Report
Comments and Suggestions for Authors
The authors have made large improvements to the manuscript in record speed. Most of the reviewer concerns have been addressed. Just a few final comments:
Could the rationale be provided for the reader as to why blocking effects are only apparent with blood normalization.
Figure 1: JQ-1 should be labeled as JQ1 to match the label in the text.
Section 2.4 and Figure S1: The mobile phase is defined, but not in any proportion, and it is mentioned that it is a gradient, but the gradient isn't described.
Figure S2: this may be a software compatibility issue, but the figure is blank for me.
Author Response
- Response to comment: Could the rationale be provided for the reader as to why blocking effects are only apparent with blood normalization.
Response: Thank you for your question. The observed increase in early brain uptake in the blocking group is likely due to unlabeled YL10 occupying binding sites in the brain, leaving more [¹¹C]YL10 in the blood pool. This higher blood activity spills over into the brain signal, particularly during the early phase, potentially obscuring the true blocking effects.
To address this, blood normalization was applied to correct for non-specific blood-derived signals, isolating the specific blocking effect of YL10. This approach clarifies the blocking results, as shown in the normalized SUV curves (Figure 6D). We have added this explanation to the Results and Discussion sections (2.8).
- Response to comment: JQ-1 should be labeled as JQ1 to match the label in the text.
Response: Thank you for pointing out this inconsistency. We have corrected the label in Figure 1 to "JQ1" to ensure consistency with the text.
- Response to comment: Section 2.4 and Figure S1: The mobile phase is defined, but not in any proportion, and it is mentioned that it is a gradient, but the gradient isn't described.
Response: We apologize for the oversight. We have now included the detailed gradient proportions and parameters in Section 2.4 and the legend of Figure S1. The gradient is described as follows:
0–1 minute: Mobile phase is 0.1% TFA in water/0.1% TFA in acetonitrile, 90/10 (v/v).
1–9 minutes: Mobile phase is 0.1% TFA in water/0.1% TFA in acetonitrile, Gradient from 90/10 to 10/90 (v/v).
9–10 minutes: Mobile phase is 0.1% TFA in water/0.1% TFA in acetonitrile, 10/90 (v/v).
These details have been updated in the corresponding sections of the supplementary material.
- Response to comment: Figure S2: this may be a software compatibility issue, but the figure is blank for me.
Response: We regret the inconvenience caused by the blank Figure S2. We have rechecked the supplementary file and confirmed that the issue was likely due to software compatibility. The updated file now includes a compatible and clearly visible version of Figure S2.

Reviewer 2 Report
Comments and Suggestions for Authors
The responses provided by the authors have markedly enhanced the quality of the manuscript. I have only a few minor comments on the revised article, which are listed below:
- Page 4, lines 132-133: the name of compound 3 is (E)-4-bromo-5-methoxy-2-((2-nitrovinyl)amino)benzoic acid and not 2-amino-4-bromo-5-methoxybenzoic acid.
- Page 4, Scheme 1: the chemical structures of compounds 8 and YL9 are erroneous. Instead of a 2-methylpyrrole or an imidazolone moiety, these compounds possess a 2-methylimidazole moiety.
- Legend of Scheme 1: please add the reagent KOAc for the conditions (7).
- Please provide the equipment used to determine the enantiomeric excess of compound 6 (see supporting information).
- In contrast with the assertions made by the authors in their responses, the enantiomeric excess of the final compound YL10, as well as that of the radiotracer [11C]YL10 (for example, determined by chiral chromatography), have not been provided. Consequently, it is recommended that the discussion section be amended to include a statement indicating that the enantiomeric excess of the reference compound and radiotracer have not been determined.
- The NMR spectrums (point E in the supplementary material file) are not displayed clearly.
Author Response
Dear Editors and Reviewer:
Thank you for your letter concerning our manuscript entitled “Development of a PET Probe Targeting Bromodomain and Extra Terminal Proteins for In Vitro and In Vivo Visualization” (Manuscript ID: pharmaceuticals-3355312). We greatly appreciate the reviewers' constructive comments, which have been invaluable in revising and improving our manuscript. We have carefully addressed each comment and made the necessary corrections, as outlined below. All revised portions are marked in red in the manuscript. The main corrections in the paper and the responds to the reviewer’s comments are as flowing:
- Response to comment: Page 4, lines 132-133: the name of compound 3 is (E)-4-bromo-5-methoxy-2-((2-nitrovinyl)amino)benzoic acid and not 2-amino-4-bromo-5-methoxybenzoic acid.
Response: Thank you for highlighting this oversight. We have corrected the name of compound 3 in the manuscript to read as (E)-4-bromo-5-methoxy-2-((2-nitrovinyl)amino)benzoic acid.
- Response to comment: Page 4, Scheme 1: the chemical structures of compounds 8 and YL9 are erroneous. Instead of a 2-methylpyrrole or an imidazolone moiety, these compounds possess a 2-methylimidazole moiety.
Response: We acknowledge the errors in the chemical structures of compounds 8 and YL9 in Scheme 1. These have been corrected, and the revised Scheme 1 now accurately depicts the 2-methylimidazole moiety.
- Response to comment: Legend of Scheme 1: please add the reagent KOAc for the conditions (7).
Response: We have updated the legend of Scheme 1 to include the reagent KOAc for the conditions (7) as requested.
- Response to comment: Please provide the equipment used to determine the enantiomeric excess of compound 6 (see supporting information).
Response: The equipment used to determine the enantiomeric excess of compound 6 has now been included in the Supplementary Material. We utilized Autopol III – Automatic Polarimeter (Rudolph Research Analytical, USA) for this analysis.
- Response to comment: In contrast with the assertions made by the authors in their responses, the enantiomeric excess of the final compound YL10, as well as that of the radiotracer [11C]YL10 (for example, determined by chiral chromatography), have not been provided. Consequently, it is recommended that the discussion section be amended to include a statement indicating that the enantiomeric excess of the reference compound and radiotracer have not been determined.
Response: Thank you for your observation. We would like to clarify that the enantiomeric excess (ee%) of both the final compound YL10 and the radiotracer [11C]YL10 has been measured. The detailed information regarding the equipment used for the enantiomeric excess determination is provided in the Supplementary Material.
- Response to comment: The NMR spectrums (point E in the supplementary material file) are not displayed clearly.
Response: We have revised and improved the clarity of the NMR spectra in the supplementary material file. The updated file now includes high-resolution spectra for better readability and analysis.
We sincerely appreciate the time and effort that the Editors and Reviewers have dedicated to improving our manuscript. We believe the revisions enhance the clarity and overall quality of the manuscript without altering its core content. Should further clarifications or modifications be required, we would be happy to address them promptly.Thank you again for considering our work for publication in Pharmaceuticals.
Sincerely,
Changning Wang, Ph.D.
Athinoula A. Martinos Center for Biomedical Imaging
Massachusetts General Hospital, Harvard Medical School
Charlestown, MA 02129, USA
cwang15@mgh.harvard.edu
